# Indirect Attention: Turning Context Misalignment into a Feature

## Abstract

The attention mechanism has become a cornerstone of modern deep learning architectures, where keys and values are typically derived from the same underlying sequence or representation. This work explores a less conventional scenario, when keys and values originate from different sequences or modalities. Specifically, we first analyze the attention mechanism's behavior under noisy value features, establishing a critical noise threshold beyond which signal degradation becomes significant. Furthermore, we model context (key-value) misalignment as an effective form of structured noise within the value features, demonstrating that the noise induced by such misalignment can substantially exceed this critical threshold, thereby compromising standard attention's efficacy. Motivated by this, we introduce Indirect Attention, a modified attention mechanism that infers relevance indirectly in scenarios with misaligned context. We evaluate the performance of Indirect Attention across a range of synthetic tasks and real-world applications, showcasing its superior ability to handle misalignment.

## 1 Introduction

The attention mechanism is a cornerstone of modern deep learning, particularly as the fundamental building block of Transformer-based architectures [1; 2; 3]. It operates by aggregating information from a context via learnable weights, effectively constructing new representations through a similarity-weighted sum of value vectors, where weights are derived from query-key dot products. This formulation inherently assumes a structural alignment in context, and keys and values originate from the same input data.

This paper asks: what happens when that assumption breaks? Specifically, we study the behavior of the attention mechanism when value vectors are drawn from a distribution different from that of keys, either due to data heterogeneity, architectural design, or multimodal inputs.

We focus on two settings: additive noise in the value vectors, and structural misalignment between the keys and values. First, we quantify the signal-to-noise ratio (SNR) of the attention output under Gaussian perturbations of the values and identify a critical noise threshold beyond which output reliability degrades. We then model key-value misalignment as an effective noise process and show that even moderate misalignment can induce noise energy far exceeding this critical threshold.

Our analysis reveals that additive noise in the values leads to an SNR decay that is independent of the model's embedding dimension. In contrast, key-value misalignment induces an SNR degradation that scales with embedding dimension, introducing irreducible noise under standard initialization assumptions. While training could, in principle, mitigate these effects, we argue that low-SNR initial states impair optimization, particularly during early training and in deeper architectures where signal degradation compounds.

Submitted to 39th Conference on Neural Information Processing Systems (NeurIPS 2025). Do not distribute.

Importantly, even when clean semantic content is available in keys, noise in the value projections can dominate and corrupt the final output.

Yet, rather than treating misalignment purely as unwanted or a bug, we argue it may hold untapped potential. Misaligned context opens possibilities for decoupling semantic retrieval (keys) from representational content (values), enabling more flexible information fusion in multimodal and cross-domain settings involving cross-modal retrieval, memory-augmented networks, or learned external context injection [4; 5; 6].

To operationalize this idea, we propose a new mechanism: Indirect Attention, designed to retain effectiveness even under misaligned contexts. We show that this architecture recovers useful behavior where standard attention fails.

We validate our theoretical insights with controlled synthetic tasks and evaluate Indirect Attention on a challenging one-shot object detection benchmark. Results demonstrate that our method maintains robustness in scenarios where conventional attention is compromised.

Our main contributions can be summarized as follows:

1. We present a theoretical analysis of attention under value perturbation, identifying a dimension-independent SNR scaling and deriving a critical noise threshold.

2. We model context misalignment as an effective noise process and show its energy typically exceeds the critical level, leading to output degradation.

3. We introduce **Indirect Attention**, a modified attention mechanism tailored to handle misaligned context.

4. We empirically validate our findings on both synthetic tasks and a real-world object detection setting concerning separate sequences.

## 2   Related works

### 2.1   Transformers robustness

Several works have considered adversarial training [7] for transformers and attention mechanisms, which encourages resistance to small, worst-case perturbations in input space. [8; 9] are using input perturbation and contrastive learning to increase the robustness of language models for text classification. Another line of work [10; 11; 12] explores adding noise to data labels as a regularization method. Our work differs from these works in the sense that we theoretically model context misalignment as noise and perturbation; otherwise, we do not aim to add noise to the model input.

### 2.2   Relative position bias

In addition to the positional encoding added to the input sequence, many works [13; 14; 15; 16; 17] use a relative position bias (RPB) that is added to attention logits based on the distance between keys and queries. However, our work differs from these works in different ways. First, in our method, we consider the relation between the queries and the value. Second, we apply the bias to the mismatched keys and values. Most importantly, the bias in our method is not merely structural and positional but gets updated at each layer by the attention layer output.

### 2.3   Cross-Attention:

Cross-attention is a foundational component of transformer-based encoder-decoder models [1] and has become central in multi-modal architectures such as [18; 5; 19; 20], where queries from one modality attend to key-value pairs from another. In such cross-attention mechanisms, the primary role of the query sequence is to extract or fuse information from the key-value sequence. Instead of a direct interaction between two sequences, indirect attention facilitates a more mediated relationship. The query sequence attends to value vectors from one context, conditioned on the key vectors from a different context. This decoupling of key and value sources enables a more flexible and robust form of information processing rather than mere fusion.

## 3 Attention under context perturbation

We begin by analyzing the behavior of the attention mechanism when the value vectors are corrupted by additive noise. This simplified case helps develop core intuitions that will later extend to misaligned or mismatched contexts.

**Notation and setup:** In the rest of the paper, mostly by *value(s)* we mean the value vectors in the attention mechanism that come along with key and query. Let $x_i \in \mathbb{R}^d$ denote the input tokens, $Q$ and $K$ denote the query and key - as the projections of $x_i$ by learned linear projections, and suppose the attention weights $a_i \in [0, 1]$ are computed using softmax:

$$a_i = \text{Softmax}(\frac{QK^T}{\sqrt{d}})_i$$

Let $W_v$ be a learned linear projection matrix applied to the input tokens $x_i$ to derive value vectors. The clean (noise-free) attention output is:

$$o^* = \sum_{i=1}^{n} a_i W_v(x_i)$$

If the input tokens $x_i$ used for the value vector construction is corrupted with additive white Gaussian noise $\epsilon_i \sim \mathcal{N}(0, \sigma^2 \mathbf{I}_d)$ such that $W_v(x_i + \epsilon_i)$, we aim to study how this noise propagates through the attention mechanism and influence the output, beginning with a bound on the deviation between clean and noisy outputs.

**Lemma 1.** *: Let $\hat{o} = \sum_i^n a_i(W_v(x_i + \epsilon_i))$ where $\epsilon_i$ is additive gaussian noise with mean 0. We denote the clean output $o^* = \sum_i^n a_i W_v(x_i)$, then the norm of the difference between the noisy and clean outputs is bounded by the weighted sum of the norms of the noise terms:*

$$\|\hat{o} - o^*\| \leq \sum_i^n a_i \|\epsilon_i\| \tag{1}$$

We leave the proof to Appendix A.1. This result, though simple, is foundational, showing that attention cannot denoise its inputs and it linearly propagates noise in the value vectors, weighted by the attention distribution, even though the clean input is given in the key vectors.

In order to gain a probabilistic understanding, we analyze the expected squared deviation between $\hat{o}$ and $o^*$ under the assumption that noise is Gaussian.

**Lemma 2.** *Let $\hat{o} = \sum_{i=1}^{n} a_i W_v(x_i + \epsilon_i)$ and $o^* = \sum_{i=1}^{n} a_i W_v(x_i)$ be the noisy and noise-free attention outputs, respectively. Assuming $\epsilon_i \sim \mathcal{N}(0, \sigma^2 \mathbf{I}_d)$ and the noise vectors $\epsilon_i$ are independent of each other and of the attention weights $a_i$, then the expected squared error of the attention output is:*

$$\mathbb{E}[\|\hat{o} - o^*\|^2] = \sigma^2 d \sum_{i=1}^{n} a_i^2 \tag{2}$$

We leave the proof to Appendix A.2. The expected output noise scales linearly with the embedding dimension $d$, and higher-dimensional representations amplify the effect of input noise. It also depends on the concentration of the attention distribution via $\sum_i a_i^2$. If the attention is sharply peaked on single point $j$ such that $a_j = 1, a_i = 0 \quad \forall \quad i \neq j$ then $E[\|\hat{o} - o^*\|^2 \approx \sigma^2 d$. On the other hand if attention is uniformly distributed $a_i = 1/n$, then $\sum_i a_i^2 = 1/n$, and the expected error is reduced to $\sigma^2 d/n$. The attention can mitigate the impact of noise when it is spread across many tokens. In particular, this result highlights a bias-variance tradeoff implicit in attention: focused attention gives high specificity but also amplifies noise from individual value vectors, and diffuse attention reduce noise but may dilute relevant information.

**Remark.** *This result extends to multi-head attention. For a model with H heads, each with dimension $d_h = d/H$, the expected squared error per head is $\mathbb{E}[\|\hat{o}^h - o^{h*}\|^2] = \sigma^2 d_h \sum_{i=1}^{n}(a_i^h)^2$. For the aggregated output after projection, assuming orthogonal rows in the output projection matrix, the expected squared error becomes $\mathbb{E}[\|\hat{o}_{MHA} - o^*_{MHA}\|^2] \approx \sigma^2 \frac{d}{H} \sum_{h=1}^{H} \sum_{i=1}^{n}(a_i^h)^2$. Thus, the core scaling behavior with respect to dimension and attention concentration remains consistent.*

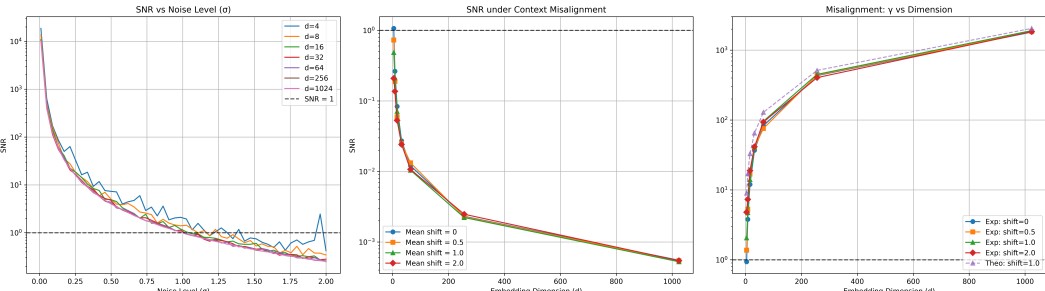

Figure 1: Analysis of attention output signal quality under noisy and misaligned contexts. Left: SNR of attention output under additive noise to value vectors remains invariant with increasing embedding dimension $d$. Middle: SNR under context misalignment degrades significantly with $d$. Right: The expected effective noise energy $\gamma$ scales with dimension and increases with mean shift between key and value distributions, matching theoretical predictions, and exceeding the critical threshold $\sigma^* = 1$.

### 3.1 Critical noise threshold: when signal equals noise

Building upon our previous analysis 2, we now shift focus to a more actionable metric: the signal-to-noise ratio (SNR) of the attention output, namely, the ratio between norms of signal and noise in the attention output. While the expected squared error quantifies absolute noise energy, SNR provides a relative measure of how much of the output is a reliable signal versus corruption.

$$SNR = \frac{\mathbb{E}[\|o^*\|^2]}{\mathbb{E}[\|\hat{o} - o^*\|^2]}$$

During training, the attention output feeds into subsequent layers and contributes to gradient computations. When the noise dominates the signal, the gradients during training become unreliable. In such scenarios, it becomes difficult for any learning mechanism, including attention, to discern the true patterns. An SNR of 1 is a natural choice as a critical threshold where, lower than that, noise dominates the attention output.

Importantly, our definition of SNR is distinct from that in works on benign overfitting or generalization with noisy labels [21]. Rather than analyzing generalization error, we focus on representation fidelity within the attention mechanism itself.

We observe that for unit-variance value vectors and additive Gaussian noise, the SNR of attention output reduces to:

$$SNR = \frac{1}{\sigma^2}$$

This gives a clean and intuitive threshold $\sigma^* = 1$, signal and noise have equal expected energy in the attention output. This critical point is independent of both the embedding dimension $d$ and the sharpness of attention weights $\{a_i\}$. In fact, under additive Gaussian perturbations, signal and noise scale identically with $d$, making their ratio purely dependent on the noise variance. Thus, independent of how large the model is, once the noise variance crosses 1, the attention output becomes dominated by noise.

**Remark.** *For multi-head attention, the SNR follows the same relation $SNR_{MHA} \approx \frac{1}{\sigma^2}$, and the critical noise threshold remains $\sigma^* = 1$ for each head. This invariance holds because both signal and noise scale proportionally with dimension across the heads.*

Figure 1 (left) shows how SNR decays as noise increases, across multiple embedding dimensions. As predicted, the critical drop to SNR = 1 occurs uniformly at $\sigma = 1$, and is unaffected by dimension.

## 4 Modeling context misalignment as effective noise

We analyze the impact of context misalignment by modeling it as an effective noise term, and compare its effect on the attention output to the critical noise threshold derived earlier. We show that the resulting error scales with the model's dimensionality and can significantly exceed this threshold.

Let $k_i = W_k(x_i)$ and $v_i = W_v(y_i)$, where in standard attention we have $x_i = y_i$, implying $v_i = W_v(x_i) = W_v(y_i)$. In the misaligned setting, we allow $y_i \neq x_i$, meaning the value vector $y_i$ is derived from a different latent source than the key such that $x_i \sim P_x(\mu_x, \Sigma_x)$, $y_i \sim P_y(\mu_y, \Sigma_y)$. We define the attention output under misalignment as $o^* = \sum_i a_i W_v(y_i)$, and the output under standard alignment as $\hat{o}^* = \sum_i a_i W_v(x_i)$. The deviation between these outputs can be interpreted as effective noise:

$$\Delta o := o^* - \hat{o}^* = \sum_i a_i(W_v(y_i) - W_v(x_i)) = \sum_i a_i \epsilon_i, \quad \text{with} \quad \epsilon_i := W_v(y_i) - W_v(x_i)$$

We define the expected squared deviation as a measure of misalignment-induced degradation:

$$\gamma := \mathbb{E}[\|\Delta o\|^2]$$

Assuming standard initialization (orthogonal $W_v$) and normalized inputs (unit variance), we can simplify the expression for $\gamma$ as:

$$\gamma = 2d + \|\mu_y - \mu_x\|^2$$

The full derivation is given in Appendix A.3. This result shows that the effective noise introduced by context misalignment grows linearly with the model's dimensionality and can easily surpass the critical noise threshold $\sigma^{*2} = 1$ identified earlier.

For example, in a model with dimensionality $d = 64$, the expected noise energy is at least 128, times higher than the critical threshold, severely compromising attention reliability under misaligned context.

**Remark.** *In multi-head attention with H heads, the effective noise scales as $\gamma_{MHA} \approx 2d + \frac{1}{H}\sum_{h=1}^{H}\|\mu_y^h - \mu_x^h\|^2$. The dimension-dependent term (2d) remains the dominant factor.*

# 5  Indirect attention

In this section, we introduce Indirect Attention, a modified attention mechanism designed to exhibit inherent robustness to misaligned context.

**Definition 1** (Indirect Attention). *Indirect attention is an attention mechanism in which the key and value vectors are derived from distinct input sequences, while each query is constructed by combining a learnable embedding with a feature from the value input. Attention scores are computed based on query-key similarity, modulated by a learnable bias function over structured relations between query and value positions. This bias evolves across layers to capture context-dependent relational patterns.*

**Mechanism description:** Let $X = [x_1, \cdots, x_n]^T \in \mathbb{R}^{n \times d}$ be an input sequence of length $n$, and let $Y = [y_1, \cdots, y_n]^T \in \mathbb{R}^{n \times d}$ be another input sequence of same length or padded to same length. Then the queries $Q = [q_1, \cdots, q_m]$ of length $m \leq n$ will be constructed from combination of learnable embeddings and a subset of size $m$ of $Y$ such that $q_i = m_i + y_{\pi(i)}$ where $m_i$ is learnable embedding and $\pi(i) \in \{1, \cdots, n\}$ selecting a position from $Y$. Then the key and value feature vectors are constructed such that $K = XW_k$, and $V = YW_v$ respectively. Let $P \in \mathbb{R}^{m \times n}$ be the relative positional index matrix defined as $P_{ij} = j - i$ capturing the offset between query position $i$ and value position $j$. Let $f : \mathbb{R} \to \mathbb{R}$ be a learnable function mapping positional offsets to attention bias vectors. Then define the biased attention score as

$$\tilde{S}_{ij} = \frac{q_i \cdot k_j + f(P_{ij})}{\sqrt{d_k}} \tag{3}$$

The indirect attention weights are computed as:

$$A_{ij}^{IA} = \frac{e^{\tilde{S}_{ij}}}{\sum_{j'=1}^{n} e^{\tilde{S}_{ij'}}} \tag{4}$$

and the attention output is

$$o_i = \sum_{j=1}^{n} A_{ij}^{IA} v_j$$

At each layer $l$, the positional matrix is updated based on a learnable function $g$ such that $p^{(l+1)} = g(o^{(l)})$ enabling the model to refine its notion of relative positions as deeper contextual features are processed. While $P_{ij}^0 = j - i$ encodes relative positions, $P_{ij}^l$ evolves to content-informed relational embedding.

Below, we discuss core details, design choices, and theoretical implications.

**Attention bias:** The attention bias serves as an addressing signal; it steers the attention mechanism toward the correct positions in the value sequence. Specifically, each query $q_i$ attends selectively to a specific position in the value sequence $v_j$. This selectivity is made possible through the learnable attention bias function $f(P_{ij})$ and the context-based updating of $P$ after each layer. Unlike standard relative position biases, our attention bias adapts dynamically to correct for context misalignment, and the positional offset matrix $P$ is contextually updated across layers to reflect latent structural shifts between queries and values. The attention bias $f(P_{ij})$ injects a structural prior into the attention mechanism. It allows the model to learn that, for a given query $i$, certain relative positions $j$ are more likely to contain relevant values. This interaction encourages each query $q_i$ to develop an implicit association with one or more likely positions in the value sequence. The model learns this alignment behavior not via explicit supervision over positions, but as a consequence of minimizing the end-task loss, where successful attention focusing leads to better predictions or reconstructions.

**Bayesian view on attention bias**: While the attention mechanism is not trained as a probabilistic model, its form resembles a log-linear model where the attention weight can be interpreted as assigning higher scores to positions based on both content similarity and structural bias. Specifically, we observe that the attention score 3 has the same additive structure as the log of a joint probability over query-key similarity and a positional prior. This motivates an interpretive analogy:

$$\log p(j|i, q_i) \propto q_i \cdot k_j + \log p(j|i)$$

where the first term behaves like a data-driven compatibility (log-likelihood) and the second term acts as a log-prior over relative positions. While this analogy is not exact, it provides useful intuition for indirect attention. We interpret the attention bias $f(P_{ij})$ as learning a log-prior over alignment positions, akin to $\log p(j|i)$. Thus by substituting $\log p(j|i)$ by $f(P_{ij})$ gives the attention weights as in 4. In practice, we use a 2-layer MLP with a ReLU activation as $f$ along the key, query, and value weight matrices, which is trained jointly with the model.

**Role of multi-head in indirect attention** The Indirect Attention mechanism naturally extends to multi-head configurations. In Multi-Head Indirect Attention, each head $h$ applies its bias function $f^h(P_{ij})$ to modulate attention weights. Different heads can develop different attention patterns—some may rely more heavily on attention bias (higher $f^h(P_{ij})$), while others focus more on content similarity (lower $f^h(P_{ij})$). See Appendix B.2.1.

# 6 Experiments

To evaluate the effectiveness of the proposed indirect-attention mechanism, we conduct two synthetic experiments and one real-world experiment. The synthetic experiments are designed to involve two separate sequences. The real-world experiment demonstrates the applicability of indirect attention in a practical scenario of one-shot object detection.

## 6.1 Synthetic Experiments

We construct tasks involving two distinct input sequences per instance. These tasks enable us to explicitly assign one sequence as the source of keys and another as the source of values, thus enforcing a controlled form of misalignment that mirrors the theoretical setup.

**Task 1: Sorting by arbitrary ordering.** This task extends the classical sequence sorting paradigm by requiring the model to sort an input sequence of letters according to a given reference ordering, rather than a fixed ( alphabetical) order. A simpler version of this task has been previously studied in

[22; 23], where sorting was performed with respect to a fixed canonical ordering. In our formulation, the model is provided with two input sequences: (1) a target sequence of letters of length 10, sampled uniformly with replacement from an alphabet of 10 symbols, and (2) a reference ordering, randomly selected from a pool of 5 distinct permutations of the alphabet. The task is to predict the index of each symbol in the target sequence such that, when sorted according to this index, the resulting sequence is consistent with the ordering constraints. The pool size of 5 reference orderings introduces generalization difficulty. While the task becomes trivial with only 2 possible orderings (nearly perfect accuracy is achievable). We generate 1000 training and 200 testing instances, each comprising a pair of input and ordering sequences.

**Task 2: Sequence retrieval.** This task requires the model to identify the location of a smaller query sequence embedded within a larger reference sequence. Like the previous task, it involves two distinct sequences serving as keys and values in the attention mechanism. Specifically, an query sequence of length 3 is sampled uniformly (with replacement) from a vocabulary of 10 numerical tokens. Independently, a reference sequence of length 10 is sampled in the same manner. The query sequence is then inserted at a randomly selected position within the reference sequence, ensuring that it is a contiguous subsequence of the reference. The model is trained to predict the starting index at which the query sequence occurs within the reference sequence. The dataset consists of 1000 training and 200 test examples, each comprising a query-reference pair.

We evaluate each model according to task-specific accuracy metrics. For the sorting task, we report per-token prediction accuracy. For the retrieval task, we report the accuracy of predicting the starting position of the input sequence within the reference sequence. We compare three transformer-based models on both tasks: one using the proposed indirect-attention mechanism, a second using a baseline attention mechanism with misaligned context (referred to as "naive misaligned attention"), and a third using cross-attention. Each model consists of 6 layers, 4 attention heads, and a hidden dimension of 128. Details of the model variants are as follows:

**Indirect Attention Transformer**: This variant replaces standard attention with the proposed Indirect Attention mechanism, where keys and values are derived from different sequences, reflecting the structure of the tasks. Specifically, keys are constructed from the context sequence (the desired ordering in the sorting task or the query in the retrieval task), while values are derived from the main content to be manipulated or located (the input to be sorted or the reference sequence). Queries are learnable embeddings that are enriched by incorporating the value sequences directly. The attention logits are modulated by an attention bias term. For the function $f$ for the attention bias, we use a two-layer MLP with ReLU activation.

**Naive Misaligned Attention Transformer**: This variant adopts a transformer-style architecture but uses conventional scaled dot-product attention directly on misaligned key and value, following the same key–value assignment as in the Indirect-Attention Transformer. The query is learned similarly.

**Cross-Attention Transformer**: This model uses a standard cross-attention mechanism where queries are derived from one sequence (the target input or query sequence) and keys and values are derived from the second sequence (the reference ordering or reference sequence).

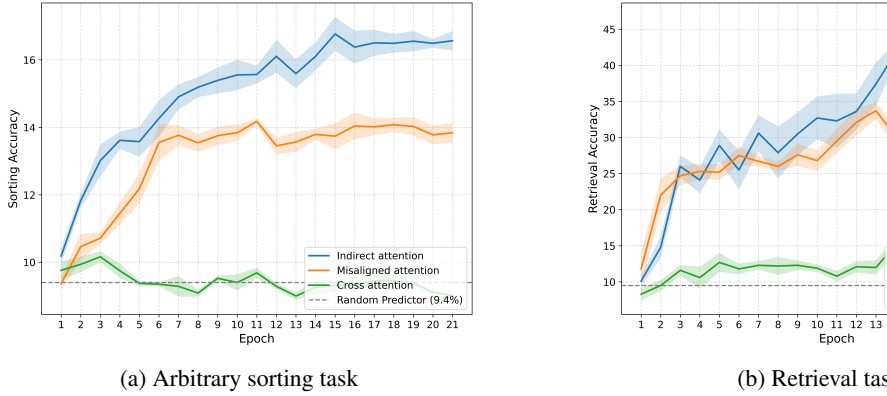

(a) Arbitrary sorting task        (b) Retrieval task

Figure 2: Comparison of test accuracy curves for three attention methods in two tasks of sorting based on given ordering and retrieval.

## 6.2 One-shot object detection

Given a training set consisting of seen classes $C_b$ and a test set containing new classes $C_n$ with $C_b \cap C_n = \emptyset$, the task of one-shot object detection is to train a detector on $C_b$ so that it can generalize to the test set and $C_n$ without additional training or tuning. Specifically, with a sample instance, also known as the query image $\mathbf{Q} \in \mathbb{R}^{H \times W \times 3}$ showing one instance of an object of a certain class, the detector is expected to display the bounding box $\mathbf{B} \in \mathbb{R}^4$ of all instances of the same class as $Q$ in the target image $\mathbf{I} \in \mathbb{R}^{H \times W \times 3}$, assuming the target image contains at least one instance of the same class as the object in $\mathbf{Q}$. This problem can also be viewed as a visual prompt task [24], where given the visual prompt $\mathbf{Q}$, the detector is expected to locate similar instances in the target image.

### 6.2.1 Experimental setup

We conduct experiments on three architecture variants on PascalVOC dataset. First, we detail the three architecture variants.

**Double cross-attention DETR:** We provide a basic setup based on DETR architecture but with necessary adaptations. Unlike DETR for normal object detection, in OSOD we deal with two separate images at the same time: the query image and the target image. To fuse both images, following [25; 26] we also add an additional block of cross-attention in the DETR architecture for aligning the target image with query image. We call this setup as double cross-attention. Given the target image features $I$ and the query image features $Q$ we have:

$$I' = \text{Cross-Attn}_1(Q, I) \tag{5}$$

$$V = \text{Cross-Attn}_2(I', O) \tag{6}$$

Where $I'$ is the aligned target image features aligned with query image by the first cross-attention block and the $V$ is the output features of second cross-attention block, and $O$ is the object queries. Each of the cross-attention blocks consists of 6 cross-attention layers.

**IA-DETR:** This variant utilizes the proposed indirect-attention. The application of the proposed indirect attention in the context of one-shot object detection is straightforward. While in previous synthetic tasks we were dealing with 1-D sequences, in here we deal with 2-D image sequences. DETR models use object queries for localizing and classifying objects, necessitating consideration of the relationships between object queries, target image features, and query image features. Instead of using two cross-attention blocks, the proposed indirect attention significantly simplifies this process by using only one indirect attention block.

| Method | Seen | Unseen |
|---|---|---|
| Double cross-atten. DETR | 77.90 | 62.31 |
| Misaligned atten. DETR | 29.21 | 33.8 |
| IA-DETR | 82.94 | 65.13 |

Table 1: The Comparisons of different architectures based on $AP_{50}$ results on PASCAL VOC.

Specifically, in the decoder, the object queries $\hat{O}^l$ are updated in the proposed indirect attention module with the consideration of both query image features $M$ and target image features $N$ as follows:

$$O^{l+1} = \text{FFN}(\hat{O}^l + \text{Indirect-Attn}(\hat{O}^l, M, N, P^l)). \tag{7}$$

In this indirect attention, the query image features $M$ serve as the key vector and the target image features $N$ serve as the value vector. This setting is important because the features are extracted from the target image features, and the query image features are used in the alignment term.

In (7), we choose to use the Box-to-pixel relative position bias (BoxRPB) [27] as $P$. The use of BoxRPB guides attention to the areas of the bounding box for each object query.

**Attention with misaligned contexts:** This model is also using only one decoder based on attention mechanism, where the context is misaligned. Specifically, the query image acts as the key, and the target image acts as the value.

The results of the comparison between the three variants are provided in the table below. The IA-DETR outperforms both other variants. The standard-attention DETR with misaligned context and values is the closest to IA-DETR in terms of architectures and number of layers, but due to misalignment between keys and values performs very poorly. The double cross-attention DETR, though benefiting from a decoder two times bigger than that of IA-DETR still can't match the performance of IA-DETR.

Table 2: Comparison results on Pascal VOC dataset. Results based on $AP_{0.5}$.

| Method | plant | sofa | tv | car | bottle | boat | chair | person | bus | train | horse | bike | dog | bird | mbike | table | Avg. | cow | sheep | cat | aero | Avg. |
|---|---|---|---|---|---|---|---|---|---|---|---|---|---|---|---|---|---|---|---|---|---|---|
| | | | | | | | | | Seen classes | | | | | | | | | | | Unseen classes | | |
| SiamFC [28] | 3.2 | 22.8 | 5.0 | 16.7 | 0.5 | 8.1 | 1.2 | 4.2 | 22.2 | 22.6 | 35.4 | 14.2 | 25.8 | 11.7 | 19.7 | 27.8 | 15.1 | 6.8 | 2.28 | 31.6 | 12.4 | 13.3 |
| SiamRPN [29] | 1.9 | 15.7 | 4.5 | 12.8 | 1.0 | 1.1 | 6.1 | 8.7 | 7.9 | 6.9 | 17.4 | 17.8 | 20.5 | 7.2 | 18.5 | 5.1 | 9.6 | 15.9 | 15.7 | 21.7 | 3.5 | 14.2 |
| OSCD [30] | 28.4 | 41.5 | 65.0 | 66.4 | 37.1 | 49.8 | 16.2 | 31.7 | 69.7 | 73.1 | 75.6 | 71.6 | 61.4 | 52.3 | 63.4 | 39.8 | 52.7 | 75.3 | 60.0 | 47.9 | 25.3 | 52.1 |
| CoAE [31] | 24.9 | 50.1 | 58.8 | 64.3 | 32.9 | 48.9 | 14.2 | 53.2 | 71.5 | 74.7 | 74.0 | 66.3 | 75.7 | 61.5 | 68.5 | 42.7 | 55.1 | 78.0 | 61.9 | 72.0 | 43.5 | 63.8 |
| AIT[32] | 46.4 | 60.5 | 68.0 | 73.6 | 49.0 | 65.1 | 26.6 | 68.2 | 82.6 | 85.4 | 82.9 | 77.1 | 82.7 | 71.8 | 75.1 | 60.0 | 67.2 | 85.5 | 72.8 | 80.4 | 50.2 | 72.2 |
| UP-DETR[33] | 46.7 | 61.2 | 75.7 | 81.5 | 54.8 | 57.0 | 44.5 | 80.7 | 74.5 | 86.8 | 79.1 | 80.3 | 80.6 | 72.0 | 70.9 | 57.8 | 69.0 | 80.9 | 71.0 | 80.4 | 59.9 | 73.1 |
| BHRL[34] | 57.5 | 49.4 | 76.8 | 80.4 | 61.2 | 58.4 | 48.1 | 83.3 | 74.3 | 87.3 | 80.1 | 81.0 | 87.2 | 73.0 | 78.8 | 38.8 | 69.7 | 81.0 | 67.9 | 86.9 | 59.3 | 73.8 |
| IA-DETR (ResNet) | 53.1 | 81.5 | 83.5 | 83.7 | 57.4 | 75.9 | 45.9 | 69.4 | 87.9 | 87.9 | 91.6 | 88.3 | 88.3 | 84 | 89.3 | 80.4 | **77.8** | 87.1 | 79.7 | 84.1 | 67.5 | 79.6 |
| IA-DETR (Swin) | 39.3 | 69.4 | 78.3 | 82.7 | 52 | 73.7 | 49.8 | 52.6 | 86.6 | 86.3 | 92.4 | 86.7 | 90.4 | 88.2 | 79.9 | 69.5 | 73.6 | 90.5 | 81.2 | 85.2 | 67.4 | **81.0** |

Table 3: Comparison results on MS COCO dataset. Results are based on $AP_{0.5}$.

| Method | Seen classes | | | | | Unseen classes | | | | |
|---|---|---|---|---|---|---|---|---|---|---|
| | split-1 | split-2 | split-3 | split-4 | Average | split-1 | split-2 | split-3 | split-4 | Average |
| SiamMask [35] | 38.9 | 37.1 | 37.8 | 36.6 | 37.6 | 15.3 | 17.6 | 17.4 | 17.0 | 16.8 |
| CoAE [31] | 42.2 | 40.2 | 39.9 | 41.3 | 40.9 | 23.4 | 23.6 | 20.5 | 20.4 | 22.0 |
| AIT [32] | 50.1 | 47.2 | 45.8 | 46.9 | 47.5 | 26.0 | 26.4 | 22.3 | 22.6 | 24.3 |
| BHRL [34] | 56.0 | 52.1 | 52.6 | 53.4 | 53.5 | 26.1 | 29.0 | 22.7 | 24.5 | 25.6 |
| IA-DETR | 53.2 | 55.6 | 56.2 | 58.1 | **55.8** | 27.3 | 27.0 | 28.7 | 26.4 | **27.3** |

### 6.2.2 Comparison with existing works:

In Table 2, we compare the performance of IA-DETR with state-of-the-art methods on the Pascal VOC dataset for both seen and unseen classes. The results clearly show that IA-DETR significantly outperforms existing methods in both categories.

To further validate the superiority of IA-DETR, we evaluate our model against other methods on the challenging COCO dataset across all four splits. The results, presented in Table 3, demonstrate that IA-DETR consistently outperforms all existing methods by an average of 2% on both seen and unseen classes.

## 7 Conclusion

We have provided an analysis of the attention mechanism's vulnerability to both context perturbation, particularly noisy value features, and the more insidious challenge of context misalignment. By formally modeling context misalignment as an effective noise, we have quantified its detrimental impact on the attention output exceeding a threshold SNR of 1. To address this challenge, we introduced Indirect Attention, a novel architectural modification specifically engineered to enhance robustness in the presence of misaligned context. Our evaluation, spanning carefully designed synthetic experiments and challenging real-world applications, demonstrates the efficacy of the proposed Indirect Attention mechanism in mitigating the adverse effects of misalignment, showcasing its potential to unlock more reliable and flexible information processing in diverse deep learning tasks.

## 8 Limitations

Our analysis in this work has primarily focused on the behavior of attention mechanisms at the initialization point, providing a foundational understanding of their inherent susceptibility to context perturbation and misalignment. However, we acknowledge that the dynamic processes of gradient descent optimization and subsequent training can induce complex and potentially mitigating behaviors that remain as an interesting direction for future work.

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

## A Proofs

### A.1 Proof of lemma 1

*Proof.*

$$\hat{o} - o^* = \sum_i a_i(W_v(x_i + \epsilon_i)) - \sum_i a_i W_v(x_i)$$

$$\hat{o} - o^* = \sum_i a_i \epsilon_i$$

Applying triangular inequality we get:

$$\|\sum_i a_i \epsilon_i\| \leq \sum_i a_i \|\epsilon_i\|$$

□

### A.2 Proof of lemma 2

*Proof.* we have $\hat{o} - o^* = \sum_i^n a_i \epsilon_i$.

$$\|\hat{o} - o^*\|^2 = \|\sum_i^n a_i \epsilon_i\|^2$$

$$\mathbb{E}[\|\hat{o} - o^*\|^2] = \sum_{i,j} a_i a_j \mathbb{E}[\epsilon_i^T \epsilon_j]$$

Since $\epsilon_i$ are iid with mean of zero we will have $\mathbb{E}[\epsilon_i^T \epsilon_j] = 0$ if $i \neq j$.

$$\mathbb{E}[\epsilon_i^T \epsilon_i] = \mathbb{E}[\|\epsilon_i\|^2] = \text{Tr}(\sigma^2 I_d) = \sigma^2 d$$

$$E[\|\hat{o} - o^*\|^2] = \sum_i^n a_i^2 \sigma^2 d = \sigma^2 d \sum_i^n a_i^2$$

□

### A.3 modeling context misalignment as additive noise

Given $x_i \sim P_x(\mu_x, \Sigma_x)$, $y_i \sim P_y(\mu_y, \Sigma_y)$ and are independent, and $\gamma = \mathbb{E}[\|\Delta o\|^2]$ we can compute the expected squared magnitude of the error $\Delta o$:

$$\gamma = \|\mathbb{E}[\Delta o]\|^2 + \text{tr}(\text{cov}(\Delta o))$$

with:

$$\gamma = W_v(\mu_y - \mu_x), \quad \text{cov}(\Delta o) = \sum_i a_i^2 W_v(\Sigma_x + \Sigma_y)W_v^T$$

Thus:

$$\gamma = \|W_v(\mu_y - \mu_x)\|^2 + \sum_i a_i^2 \text{tr}(W_v(\Sigma_x + \Sigma_y)W_v^T)$$

To simplify the subsequent analysis, we introduce two commonly used approximations:

1. Orthogonal projection weights: Assume $W_v$ has orthogonal rows with unit singular values (as motivated by common initialization schemes [36; 37]).

2. Normalized inputs: Inputs are normalized such that each component has unit variance.

Under these approximations:

$$\text{tr}(W_v \Sigma_x W_v^T) \approx \text{tr}(\Sigma_x) = d \quad \text{and} \quad \|W_v \mu\|^2 \approx \|\mu\|^2$$

Additionally, the squared norm of the aligned output is approximated as:

$$\|o^*\|^2 \approx \|\mathbb{E}[o^*]\|^2 + \sum_i a_i^2 \operatorname{tr}(W_v \Sigma_x W_v^T) \approx \|\mu_x\|^2 + d \sum_i a_i^2$$

Substituting the above into $\gamma$ we get:

$$\gamma = \|\mu_y - \mu_x\|^2 + 2d$$

## A.4 Key and value assignment in indirect attention

We follow the same consistent assignment across all tasks for query, key and value:

**Keys:** are constructed from the conditioning sequence.

**Values:** are constructed from the content sequence (i.e., the sequence over which we expect attention output to be formed).

**Queries:** are learned embeddings, optionally enriched with features from the value sequence.

## A.5 Visualization

For the retrieval task, we visualize the attention bias for each layer and attention head in 3. The attention bias for the first layer is just a position bias, but for the subsequent layers, it considers the data as well.

# B IA-DETR

## B.1 Architecture

The IA-DETR architecture and a comparative figure showing how indirect attention leads to a more efficient architecture are shown in 4 and 5.

## B.2 Implementation detail

For the backbone, Swin Transformer pre-trained on ImageNet with MIM is selected for the proposed approaches. For IA-DETR, a backbone of ResNet-50 pre-trained on reduced ImageNet is also tested. For the first stage of training, we train the model for 30 epochs with a batch size of 24 on 4 GPUs using the SGD optimizer. In this stage, we keep the backbone frozen and only train the decoder part. In the second stage, we train the model for 14 epochs with a batch size of 16 on 8 GPUs using the SGD optimizer. We follow the works [32; 34; 31] to generate the target-query image pairs. During training, for a given target image containing an object from a seen class, we randomly select a patch of the same class from a different image. During testing, for each class in the target image, query patches of the same class are shuffled using a random seed set to the image ID of the target image. The first five patches are then selected, and the average metric score is reported.

### B.2.1 Visualization

To comprehend the behaviors of indirect attention, we conducted extensive visualization of the attention maps. Through our analysis, we made the following observations as expected.

- Certain attention heads prioritize the content of the query image features, while others concentrate on specific locations within the target image features efficiently disentangling the alignment term from the positional bias.
- The indirect attention mechanism selects object queries based on the conditioning provided by the query image features.

To investigate how queries are ranked in the alignment term, purely for visualization purposes, we compute the output of the dot product between the query and key. Although the model applies softmax along the key dimension, for query ranking understanding, we reverse the softmax operation by performing softmax along the query dimension. It's noteworthy that not all attention heads focus

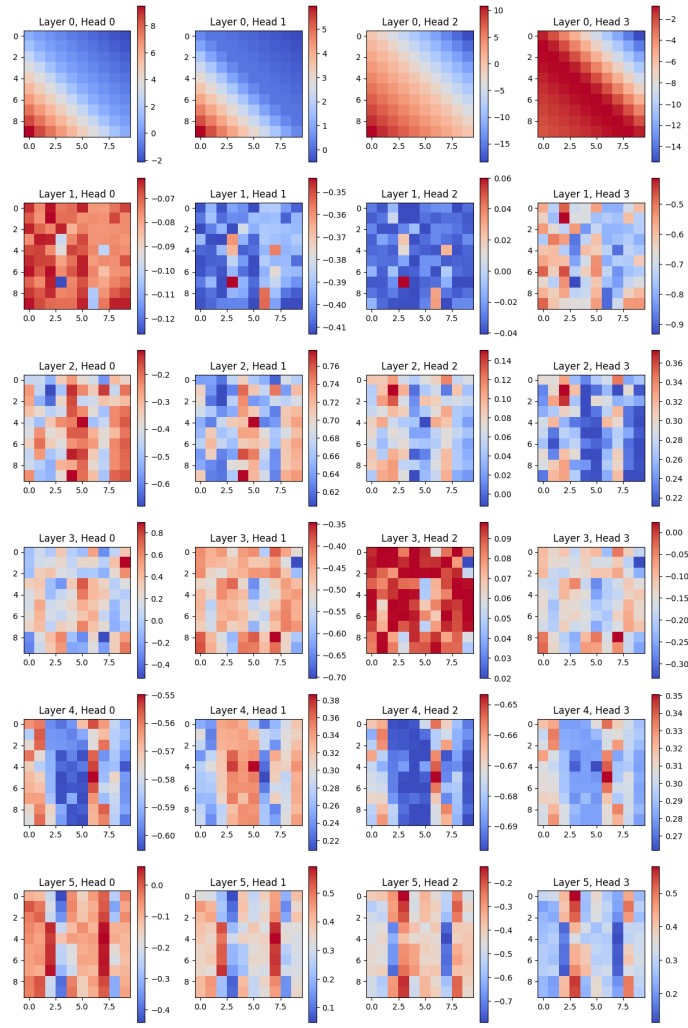

Figure 3: Attention bias for each layer and each attention head for the retrieval task.

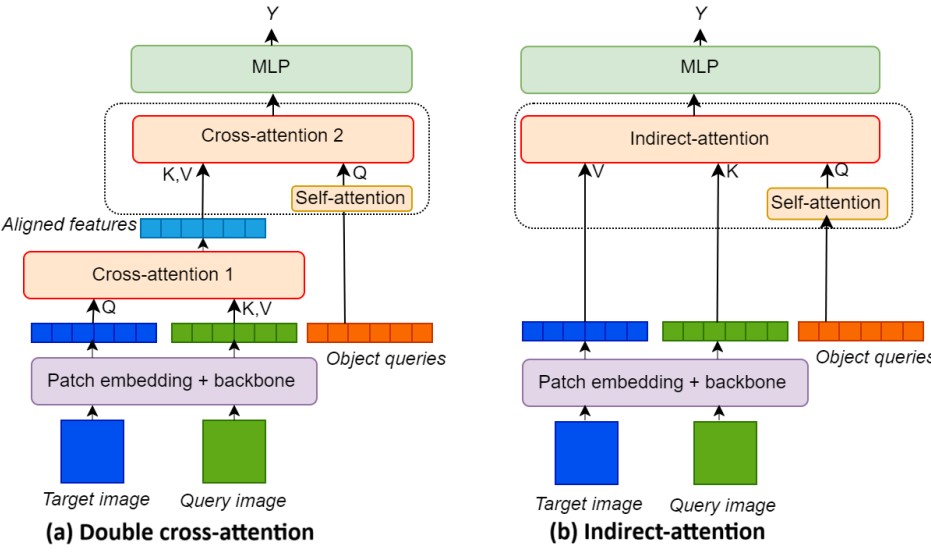

Figure 4: Comparison between double cross-attention and Indirect-Attention for OSOD.

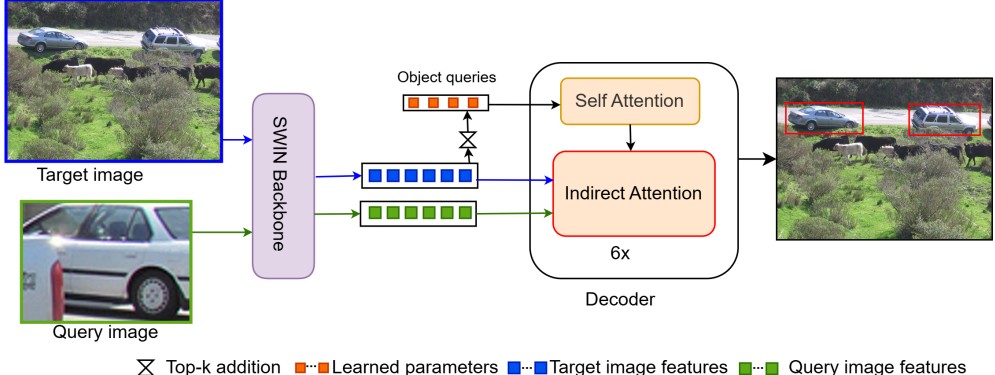

X Top-k addition  ■···■ Learned parameters  ■···■ Target image features  ■···■ Query image features

Figure 5: A workflow of IA-DETR architecture for one-shot object detection. The target and query image features coming from two distinct images are used as value and key in indirect attention.

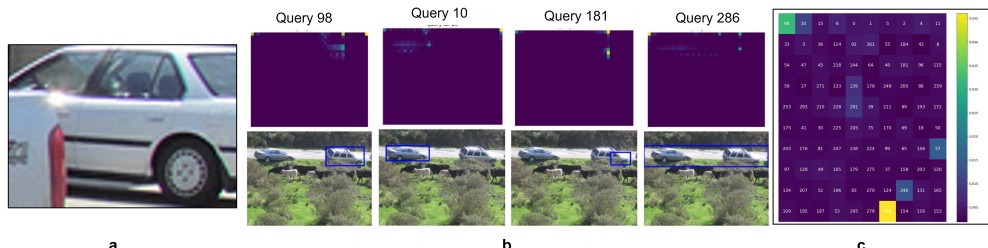

Figure 6: (a): Query image. (b): attention map from positional bias heads and detected objects of 4 object queries. (c): attention score from semantic alignment heads based on object queries. Number in each cell shows the query id.

on the alignment term, only specific heads do. We extract values along these specific heads, averaging them. Then, we average again along the key dimension, resulting in a vector with the same length as the number of object queries. The values of this vector are well aligned with the model's confidence score in the final prediction, and the object queries similar to the query image show higher values. To enhance visualization, we reshape the ordered vector into a two-dimensional matrix. As illustrated in Figure 6, object queries related to the query image object receive higher attention than others. In fig. 6 we visualize attention maps and scores for both semantic alignment and structural alignment separately.

