# OpenReview forum: "Indirect Attention: Turning Context Misalignment into a Feature"
_NeurIPS.cc/2025/Conference — Submitted to NeurIPS 2025_

### Official Review · Reviewer_4RHD · 2025-06-26

**Clarity:** 3
**Significance:** 3
**Originality:** 3
**Rating:** 4
**Confidence:** 3

**Summary:**

This paper focus on the attention mechanism in the case of keys and values generated from different input sequence. The author analyze the attention mechanism with a noised value and establish a critical noise threshold for signal degradation. Then, they model a context misalignment to prove this misalignment causing attention's efficacy. Based on the motivation above,  this work proposed indirect attention with a learnable  position bias. The experiments are conducted on synthetic tasks and real-world one-shot object detection.

**Questions:**

In addition to the above weakness, there are some questions:

Q1. Correlationship between attention bias and semantic information: I think the attention bias in Layer $i (i>=1)$ is lack of interpretability, it is difficult to determine whether the improvement in model performance is due to the additional parameters or the designed mechanism.

Q2. Line 98, Line 467, why $W_v$ is neglected?

Q3. As the size of the attention score matrix is $N_q \times N_k$, which is extremely large in the case of long sequence input, such as long context, high resolution image and long video, the bias matrix and the $f$ function might introduce lots of additional computation causing low efficiency in both speed and memory. Please provide an analysis on efficiency or the balance between efficiency and performance.

If the author solves the question and weakness I raised, I will increase the rate accordingly.

**Ethical Concerns:**

["NO or VERY MINOR ethics concerns only"]

**Final Justification:**

In the rebuttal session, authors have addressed all my concerns raised above. All these additional information in rebuttal would be useful with added clarity for this work, thus being worthwhile to be part of it. In my assessment, this work satisfies the acceptance criteria of NeurIPS and I will increase my rate to 4 (Borderline accept).

**Limitations:**

This work only focuses on the behavior of the attention mechanism during initialization, providing fundamental insights into its inherent sensitivity to contextual perturbations and misalignments. However, the dynamics of gradient descent optimization and subsequent training may induce complex and potentially mitigating behaviors. Additionally, the assumption in the theoretical analysis is simple, which weaken universality of theoretical conclusions. The application scope of the model is extended to the feasibility of tasks such as syntactic misalignment in cross-language translation and temporal misalignment between video and text.

**Paper Formatting Concerns:**

There are no obvious formatting problems with this work.

**Quality:**

3

**Strengths And Weaknesses:**

**Strengths**:

S1. This paper is well organized with good writing.

S2. Solid Theory:  The paper proposes rigorous theoretical analysis, quantifying the impact of the value with additive noise and key-value misalignment on attention output.

S3. The experiments show promising improvement comparing with baseline.

**Weaknesses**:

W1. The baselines used in the experiments of this paper are outdated. It is suggested to cite some recent works as baselines.

W2. As the $P^l$ is updated with $l$-th layer, more analysis about the initialization of $P^0$ and $f$ is needed.

W3. Line 216, do the authors make some ablations on  $f$ ? Why use MLP? The MLP might introduce more computation.

---

> ### Author Rebuttal · Authors · 2025-07-28
>
> We appreciate that reviewer 4RHD finds the paper clear and well-written, the theory solid, and results promising. We thank the reviewer for the careful review and thoughtful comments and feedback. Below, we provide response to each of the raised concerns and questions one by one.
>
> **W1:** Thanks for mentioning this, we have expanded the comparisons table for OSOD tasks in the revised version. We had missed "Semantic-aligned fusion transformer for one-shot object detection (CVPR, 2022)". We still outperform the mentioned work by average of 5% on Pascal VOC unseen, COCO seen and unseen classes, but we underperform it only in Pascal VOC seen classes by 2%.
>
> **W2 and W3:** As both weaknesses are related to relative positional matrix ($P$) and attention bias function($f$), let us address both together. In our experiments, we initialize $P^0_{ij}$ as the relative position $j - i$. This choice provides a simple inductive prior that encodes symmetric distance information.
>
> Regarding $f$, in our initial experiments, we considered a simpler form for the attention bias $f(P_{ij})$, where it was an identity function that merely broadcasted the raw relative position $P_{ij}$ over attention heads.
> Compared to this "identity function" approach, employing a lightweight 2-layer MLP for $f(P_{ij})$ significantly improved performance.
> Regarding the computational cost, let us provide extensive details in Q3 as it is relevant.
>
>
>
> **Q1. explainability of the correlationship between attention bias and semantic information:**
>
> Initially, $f(Pij​)$ operates on a purely structural, relative positional offset $(P_{ij}​=j−i)$. In early layers, its learning is primarily driven by capturing common positional patterns that lead to relevant information. We agree that in the deeper layers, the interpretability becomes more complex *(as with nearly all deep learning approaches)*, as it's no longer just about raw positions but about how the model learns to relate semantically enriched positional features to attention scores.
> Using an MLP for $f$ indeed introduces additional parameters. However, if it were just about additional parameters, the Double cross-attention model has more layers and more parameters than the indirect-attention-based model (we provide a comparison of the number of parameters in response to question 3).
>
>
> **Q2: Neglection of $W_v$**
>
> Thank you for mentioning this, we had forgotten to mention the assumtion of orthogonal at initialization for $W_v$, we have rectified it. The $W_v $ is neglected in these derivations under the common assumption that it is approximately orthogonal at initialization for convenience and tractability, which holds in practice with some initialization methods. This allows norm preservation: $||W_v \epsilon_i|| = ||\epsilon_i|| $. However, in our experiments, we do not limit ourselves to this initialization method.
>
> **Q3: computation complexity**
>
> We provide the computation complexity comparison indirect-attention based model and the double cross-attention model for the OSOD task for low and high resolution images. We do not consider the misaligned self-attention as it performs very poorly.
>
> | Method | Image Size | FLOPs (G) | Memory (GB) | # Params.
> |---|---|---|---|---|
> | Double Cross-Attention | 512x640 | 186.3 | 9.7 | 69M |
> | | 1024x1024 | 536.3 | 26.7 | 69M |
> | Indirect Attention | 512x640 | 173.7 | 9.4 | 61M |
> | | 1024x1024 | 478.2 | 23.3 | 61M |
>
> The problem of computational complexity is inherent to attention mechanisms. However, $f$ (just a linear projection)  adds minimal complexity compared to the full attention matrix multiplication (one major source of computation complexity in attention mechanisms).
>
> Regarding performance, the COCO dataset already contains images of varying sizes and aspect ratios, so the evaluation result provided in table 1 is general regarding the image resolution.

---

> > ### Comment · Reviewer_4RHD · 2025-08-01
> > **post rebuttal interaction**
> >
> > Thank you to the authors for providing additional explanations and experiments addressing my concerns—these have been greatly helpful. All this information should be incorporated into the paper and its appendix.
> > Additionally, I believe this idea could be extended to linear attention or recurrent models, as their state update mechanisms are analogous to the layer update process described in this paper. It would be intriguing to combine the proposed method with other attention-based models such as linear attention variants (e.g., PolaFormer [1] and GLA [2]) and SSMs (e.g., Mamba [3], VMamba [4]), even though these are single-modality models.
> > Mentioning these directions and cite related works in the paper’s future work section would be valuable and of long-term significance.
> >
> > [1] Meng, Weikang, et al. "PolaFormer: Polarity-aware linear attention for vision transformers." ICLR, 2025.
> >
> > [2] Yang, Songlin, et al. "Gated linear attention transformers with hardware-efficient training." arXiv preprint arXiv:2312.06635 (2023).
> >
> > [3] Dao, Tri, and Albert Gu. "Transformers are ssms: Generalized models and efficient algorithms through structured state space duality." arXiv preprint arXiv:2405.21060 (2024).
> >
> > [4] Liu, Yue, et al. "Vmamba: Visual state space model." Advances in neural information processing systems 37 (2024): 103031-103063.

---

> ### Author Response · Authors · 2025-08-01
>
> Thanks to the reviewer's comments and feedback, we have already made rectifications and expanded in both the paper and appendix. We are reviewing the referenced works and incorporating them into the final manuscript. The connection to state space models and linear attention mechanisms presents an interesting direction for future research.

---

> > ### Comment · Reviewer_4RHD · 2025-08-01
> > **post rebuttal interaction 2**
> >
> > Thank the authors for addressing all my concerns. All these would be useful with added clarity for this work, thus being worthwhile to be part of it. In my assessment, this work satisfies the acceptance criteria of NeurIPS and I will increase rating my rate accordingly.

---

### Official Review · Reviewer_vSoh · 2025-06-30

**Clarity:** 3
**Significance:** 3
**Originality:** 4
**Rating:** 5
**Confidence:** 4

**Summary:**

The standard scaled-dot-product attention assumes that keys and values come from the same distribution.
This paper explores what happens when that assumption is violated. The paper initially provides a theoretical setup showing how additive noise emerged from a different distribution of value vectors propagates into the model, perturbs the model output. Based on this paper, views this context misalignment as an effective noise that scales linearly with the embedding dimension, often far exceeding a critical noise threshold. Subsequently, the paper introduces Indirect Attention, a variant in which queries attend to values from one sequence conditioned on keys from another, augmented with a learnable, content-adaptive positional bias. The positional bias is updated layer-by-layer to make IA more robust than standard misaligned attention and cross-attention mechanisms. While the paper addresses a critically important and an overlooked problem in modern transformer based architecture, the paper lacks qualitative validation on real world tasks.
Overall, the paper is well written, informative and proposes a novel mechanism.

**Questions:**

- How sensitive is the performance of IA to the capacity of the bias function f and to updating the positional matrix layer by layer?
- Does IA provide any computational advantage of cross-attention or standard attention?
- How would IA performance be on segmentation tasks where spatial precision matters? Would IA perturb the  fine boundaries?
- Modern detectors uses features at different scales, can IA be safely used to mix key and values from different scale?
- For multimodal models, can the same theoretical guidance of misalignment to effective noise be applied?
- When you freeze the keys and optimize only values, the bias function, and projection matrix, can IA offer a cheaper task adaptation than fullfinetuning?
-  In IA-DETR architecture shown in *Figure 5*, at which stage is the positional bias injected?
- SNR = 1/σ²  in  lines 137-143  Have you assumed both value and additive noise have  identical variance?

**Ethical Concerns:**

["NO or VERY MINOR ethics concerns only"]

**Final Justification:**

Authors have addressed most of the concerns raised by the reviewers properly

**Limitations:**

Yes

**Quality:**

3

**Strengths And Weaknesses:**

**Strengths**
- The motivation of the paper is quite clear and well established. The authors provided a comprehensive theoretical analysis of the research question and extending it to multi-head attention mechanism. The theory is validated empirically with toy and real tasks, in which the proposed Indirect Attention outperformed the standard attention mechanism.


- Though the research shares similar notion of cross attention, the give method recasted misalignment as a structured noise, a novel mechanism differ from previous works which can work with both unimodals and multimodal models.


- The paper is well written and structured;notation, lemmas and assumptions are explicit; figures effectively convey the key insights of the idea.

**Weaknesses**
- The proof of modelling context misalignment relies on orthogonal initialization and unit-varainace inputs. The paper only provide the effect of training time dynamic qualitatively.

- In addition to equations,  a simple diagram explaining the IA mechanism compared to standard attention would be helpful.

- Paper lack of real data to validate the claims. Authors could explore some simple tasks on NLP domain to further validate the benefits of the proposed method.
 - A theoretical explanation on how IA reduced the effective noise comes from misaligned context would be helpful.

---

> ### Author Rebuttal · Authors · 2025-07-28
>
> We appreciate that reviewer vSoh finds our work clear and wellwritten, sees the theoretical analysis as comprehensive, and finds the proposed method novel. We thank the reviewer for the detailed review and valuable feedback and comments. We provide response to the raised questions and concerns in below.
>
> **W1:** Indeed, our theoretical analysis relies on orthogonal initialization of $W_v$ and unit-variance inputs. These choices are not only common for analytical tractability in theoretical deep learning studies but also reflect some common practices in initializing models, where weight matrices are orthogonalized and inputs are layer-normalized to unit variance at each layer or the input data is normalized at the beginning of the model.
>
>
> **W2:** Thanks for mentioning this we have expanded on this in our revised version for the final version.
>
> **W3:** We agree that NLP tasks would complement the experiments, but as the authors’ main specialization has been more towards computer vision, we have tried to focus on CV task. However, since the theoretical analysis does not depend on a specific data type, it should be applicable to NLP tasks as well.
>
>
>
> **Q1:** Regarding the bias function $f$, during our experiments we observed that moving from a fixed function to a shallow MLP leads to notable gains, but deeper networks did not yield too much improvements. A moderate capacity in $f$ is sufficient to capture useful positional priors without overparameterization and increased computation complexity.
> Regarding the layer-wise update of the positional matrix $P$, similar to normal DETR models we found that updating it dynamically (i.e., $P^{l+1} = g(o^{(l)})$) leads to better alignment and performance. Overall a depth of 4 layers in synthetic tasks and normal depth of 6 layers (standard for DETR models) gives good balance between performance and computational complexity.
>
> **Q2:** Yes, we provide the benchmark of IA-detr compared to double-cross attention variant in the following. However the standard attention on the OSOD task performs very poorly (as shown in table 1) so we do not consider it in this comparison.
>
> | Method | Image Size | FLOPs (G) | Memory (GB) | # Params|
> |---|---|---|---|---|
> | Double Cross-Attention | 512x640 | 186.3 | 9.7 | 69M |
> | | 1024x1024 | 536.3 | 26.7 | 69M |
> | Indirect Attention | 512x640 | 173.7 | 9.4 | 61M |
> | | 1024x1024 | 478.2 | 23.3 | 61M |
>
>
> **Q2:** We have mainly focused on object detection, but since in the segmentation, mainly the detection head is being replaced by a segmentation head, we expect it to not affect the performance on a segmentation task much.
>
> **Q3:** In fact in current implementation though not multi-scale extracted from backbone, but we already use mixed multi-scale by simple resizing of backbone’s last layer feature output. So the method already handles mixed multi-scale features.
>
> **Q4.** Our theoretical analysis does not depend on any specific type of data and we assume the keys and values coming from two different distributions which are well aligned with multi-modal data.
>
> **Q5:** If we understood the question correctly, normally in a one shot object detection task, there is no fine-tuning at all. So once the model is trained on seen classes we have directly tested on unseen classes without any further fine-tuning.
>
> **Q6:** In figure 5, the position bias is applied in the Indirect Attention module, but because it clutters the figure we have avoided to depict it in detail.
>
> **Q7:** No, we do not assume the signal and noise share the same variance. We assume unit-variance signals and the noise $\epsilon_i$ are sampled i.i.d. from $\mathcal{N}(0, \sigma^2 I_d)$.

---

> ### Comment · Reviewer_vSoh · 2025-08-05
>
> Thank you to the authors for thoroughly addressing all of my concerns. The clarifications and additional insights provided—particularly around the theoretical assumptions, implementation details, and generalization across tasks are helpful. I believe incorporating these points will further strengthen the paper. In my assessment, this work meets the acceptance criteria for NeurIPS, and I will update my rating accordingly.

---

### Official Review · Reviewer_oBvC · 2025-07-03

**Clarity:** 1
**Significance:** 2
**Originality:** 2
**Rating:** 3
**Confidence:** 4

**Summary:**

This paper investigates attention mechanisms under the condition of key-value misalignment, specifically when queries and values originate from the same distribution while keys come from a different one. The authors theoretically and empirically demonstrate that noise in values is preserved rather than suppressed in standard attention when such misalignment occurs. To address this, they propose indirect attention, which introduces an adaptive attention bias to correct for content misalignment. The effectiveness of this approach is evaluated through synthetic tasks and a real-world one-shot object detection scenario.

**Questions:**

1. Clarify the Motivation and Distinction from Cross-Attention:
Could the authors clarify what specific scenarios motivate studying this particular setup, and when does this misalignment naturally arise in real-world applications?

2. Indirect Attention under Self-Attention:
While the paper focuses on misaligned attention in cross-source contexts, how would the proposed indirect attention behave under self-attention settings? Could the authors clarify if and how the method generalizes to standard transformer blocks?

3. Experimental Fairness and Ablation Study:
Could the authors provide an ablation study or a fairer comparison for the OSOD experiment?

4. Missing Related Work:
The paper does not cite or discuss “Alignment Attention by Matching Key and Query Distributions” (NeurIPS’21), which is highly relevant to the topic of distributional alignment in attention. A comparative discussion of how indirect attention differs from or improves upon such work would help place the contribution in context.

**Ethical Concerns:**

["NO or VERY MINOR ethics concerns only"]

**Limitations:**

yes

**Quality:**

2

**Strengths And Weaknesses:**

Strenghs:
- This paper addresses a less-studied but potentially important case of the attention mechanism misalignment between keys and values.
-  The authors propose a novel variant, indirect attention, which introduces adaptive attention bias as a potential fix.
- They demonstrate effects of misalignment through controlled synthetic  and real-world cases.

Weakness:
- The motivation and real-world applicability of the misalignment setting remain unclear.
- The distinction between the proposed setting and standard cross-attention is not well justified.
- Experimental results are limited: only one real-world task is tested, and comparisons are made with weaker baselines.
- Ablation studies in the real-world experiment are missing, so it is unclear whether the improvements come from indirect attention itself or from architectural differences.
- Relevant prior work, including Alignment Attention by Matching Key and Query Distributions (NeurIPS'21), is not cited or discussed.
- Writing and organization are weak in several parts, reducing clarity.

This paper tackles an underexplored failure mode in attention mechanisms—key-value misalignment—by analyzing cases where queries and values come from the same distribution but keys come from another. The proposed indirect attention introduces adaptive bias to mitigate the effects of such misalignment. The problem is interesting and the synthetic experiments help illustrate the issue, while the method itself is intuitively motivated. However, the paper falls short in several key areas. First, the practical motivation is not clearly established: it's unclear when such misalignment naturally occurs, and why it matters in real-world scenarios. Furthermore, the introduction claims that decoupling semantic retrieval (keys) from representational content (values) is beneficial, but this is neither elaborated nor supported experimentally. The real-world evaluation is limited to one-shot object detection with weak baselines, and the improvements shown may stem from architectural differences rather than the proposed attention mechanism itself, as no ablation studies are provided. Notably, the paper omits discussion of relevant prior work, including “Alignment Attention by Matching Key and Query Distributions” (NeurIPS’21), which directly addresses distributional alignment in attention. Writing-wise, the introduction lacks cohesion and contains abrupt shifts. The discussion of transformer robustness in Section 2.1 feels disconnected, and the mention of position bias later on is unmotivated. Overall, while the paper raises a compelling question and proposes a potentially useful mechanism, the lack of clarity in motivation, limited experimental evidence, and weak connection to existing work undermine its impact.

---

> ### Author Rebuttal · Authors · 2025-07-28
>
> We appreciate that reviewer oBvC sees the potential importance of our work, and finds the method novel. We also thank the reviewer for the thoughtful comments, feedback, and constructive advise. The concerns raised are important and we provide responses to raised weaknesses and questions as follows.
>
> **Weaknesses:**
>
> **Motivation and natural occurrence of key-value misalignment:**
> Indeed, explicit key-value misalignment has not been studied in prior literature and at first view it seems more like a bug, and unnatural. Our contribution is to explore this and turn it into a feature. While the setup is intentionally non-standard, we believe it opens avenues for architectures that are more robust or expressive. In existing works such misalignment occurs implicitly in the shape that given two sequences of $A$ and $B$, first they are aligned using a block of cross-attentions to get aligned features $C$, and then a second seperate block of cross-attentions between learnt query and $C$ is used to extract something specific from $C$. In addition to the one-shot detection task provided in paper, this scenario appears implicitly in following works [1, 2, 3].
> In the one-shot detection task we show that how the proposed indirect-attention can replace the two blocks of cross-attention while achieving better performance with less number of parameters and less computational complexity. Moreover, it should be noted that query and value are not from the same data distribution in the proposed indirect-attention and query is mainly learnable parameters and even not the same length as the value sequence.
>
> **Distinction from Cross-Attention:**
> Section 2.3 puts the effort to distinguish our method from standard cross-attention and we accept that the difference is more naunce and difficult to articulate, and to clarify further in simple words: in indirect attention, all three components of queries, keys, and values are drawn from distinct sources with the query being learnable while in cross-attention there are two distinc sources. Moreover, it can be seen in the synthetic tasks the cross attention performs poorly similar to random predictor.
>
> **The scope of real-world evaluation and baselines:**
> We chose one-shot object detection (OSOD) as a representative application where three seperate sequences of object queries, reference image, and query image requires a double cross-attention setup. While OSOD is not the only domain where such misalignment is relevant, it offers an easy setting for introduction of the concept and empirical study. We have included all recent baselines in this area, except Semantic-aligned Fusion Transformer (CVPR 2022), which we acknowledge and will incorporate in the final version. Notably, our method still outperforms it on both COCO and PASCAL datasets by large margin.
>
> ___
> **Questions:**
>
> **Q1:**  Thank you for raising this point. In the revised version prepared for the final submission, we have included a simple figure illustrating the differences between indirect attention (IA), cross-attention, and self-attention.
>
> Regarding the scenarios where IA is applicable, it is particularly well-suited for tasks where predictions over a sequence must be conditioned on another sequence. For example, in arbitrary sorting, the model must sort a sequence not according to a memorized alphabetical or numerical order, but based on a custom ordering provided at inference time, such as ordering [b, c, a, d] where 'b' precedes 'a'.
>
> Similarly, in the One-Shot Object Detection (OSOD) setting, the goal is to detect objects in a target image that belong to the same class as a query image. Cross-attention is less effective in such cases because it aligns and fuses two sequences. While cross-attention can be used in OSOD by introducing an additional cross-attention block as presented in the paper. This adds complexity and computational overhead. In contrast, our paper demonstrates that IA, even with a single attention block, can achieve superior performance.
>
> **Q2:** In fact, our suggested Indirect Attention is specifically designed for the case of misaligned context and can be easily used in standard transformer blocks when facing scenarios explained in Q1.
>
> **Q3:** We provide the following ablation on the impacts of  positional bias $f(P)$ and positional bias matrix updates by semantic information at each layer as two main components of IA:
> | Method | Seen classes | Unseen classes |
> |---|---|---|
> | W/O positional bias | 34.3 | 35.27 |
> |W/O positional bias update | 79.8 | 69.3 |
> | Indirect Attention | 81 | 73.6 |
>
>
>
> **Q4 Citing 'alignment attention' paper:** Thanks for mentioning the “alignment attention” paper, it is relevant to our line of work and we have included and expanded on it in the related work section in our revised version for final version.
>
> [1]: Fs-detr: Few-shot detection transformer with prompting and without re-training (ICCV, 2023)
>
> [2]: BLIP: Bootstrapping language-image pre-training for unified vision-language understanding and generation (ICML, 2022)
>
> [3]: Align before fuse: vsion and language representation learning with momentum distillation (NeurIPS, 2021)

---

### Official Review · Reviewer_qjrB · 2025-07-03

**Clarity:** 4
**Significance:** 3
**Originality:** 4
**Rating:** 4
**Confidence:** 3

**Summary:**

This paper investigates how attention mechanisms degrade when keys and values come from misaligned contexts, such as different sequences or modalities. The authors provide a theoretical analysis showing that key-value misalignment acts as structured noise, often exceeding a critical signal-to-noise threshold and leading to unreliable outputs. To address this, they propose Indirect Attention, a mechanism that decouples keys and values and introduces a learnable attention bias that adapts across layers. They evaluate the method on synthetic sorting and retrieval tasks, as well as one-shot object detection benchmarks, showing improved robustness over standard and cross-attention baselines.

**Questions:**

1. Can you provide ablations isolating the effect of the attention bias function and the dynamic update of the positional index matrix to better understand which components of Indirect Attention are driving performance?

2. How sensitive is the model to the assumption of orthogonal initialization and unit variance inputs in practice? Have you observed any degradation when these assumptions are relaxed during training?

3. The synthetic tasks seem tailored to highlight the benefits of Indirect Attention under controlled misalignment. Can you evaluate the method on more challenging or naturalistic misalignment settings to better assess generality? I acknowledge this might be a bit of a high-level comment but I am extremely interested in how this would play out.

4. In the one-shot detection experiments, can you clarify whether the performance gains over DETR are due primarily to the alignment mechanism or architectural simplifications (e.g., fewer decoder blocks)? A controlled comparison would help.

5. Since the paper argues that low-SNR initialization can hinder optimization, can you provide empirical evidence (e.g., training curves, gradient norms, or convergence speed) showing how Indirect Attention mitigates this issue during early training compared to standard or cross-attention?

**Ethical Concerns:**

["NO or VERY MINOR ethics concerns only"]

**Final Justification:**

After rebuttal and discussion, I’m keeping my borderline accept recommendation. The authors added ablations showing the roles of the positional bias and dynamic index updates, clarified that their results don’t rely on orthogonal initialization or strict input normalization, and argued that one-shot detection already captures a natural misalignment scenario. They also gave some evidence that Indirect Attention converges faster under noisy values. These additions address some of my earlier questions, but the evaluation still feels narrow, with synthetic tasks tailored to the method and limited testing on more varied misalignment cases. The theoretical framing and simplicity remain strong points, but the scope of empirical validation still limits my confidence in broader applicability.

**Limitations:**

The paper includes a brief limitations section acknowledging that the analysis focuses on initialization and does not account for how training dynamics may mitigate misalignment. It would be helpful if the authors could expand this section to discuss potential failure modes during optimization or deployment, and to consider whether there are contexts where misalignment may not be harmful or could be handled more easily through training.

**Paper Formatting Concerns:**

n/a.

**Quality:**

3

**Strengths And Weaknesses:**

*Strengths:*

1. The paper provides a clear and well-scoped theoretical analysis of how additive noise and key-value misalignment degrade attention outputs, introducing a clean SNR-based threshold that grounds the problem.

2. The formulation of context misalignment as structured noise is both intuitive and technically sound, and the derivation showing how its impact scales with embedding dimension adds depth to the argument.

3. The proposed Indirect Attention mechanism is simple, generalizable, and integrates a learnable bias over key-value positions in a way that adapts across layers, which is a reasonable architectural modification.

4. The empirical results are consistent across both synthetic and real-world tasks, and the one-shot detection results show competitive gains over stronger baselines despite using fewer parameters.

*Weaknesses:*

1. The synthetic tasks are constructed to heavily favor the proposed setup and are too simplistic to support broader claims about generalization under arbitrary misalignment.

2. The paper lacks ablation studies isolating the contribution of the dynamic attention bias or the layerwise updating of position indices, which makes it hard to attribute gains to specific design choices.

3. The theoretical analysis relies on assumptions like orthogonal initialization and normalized inputs, but the paper doesn’t test how sensitive the method is to violations of these assumptions in practice.

---

> ### Author Rebuttal · Authors · 2025-07-28
>
> We appreciate that reviewer qjrB finds our theoretical analysis clear and intuitive, our proposed method simple and generalizable, and the results consistent. We thank the reviewer for thoughtful comments and feedback, and interest in the paper. Below we provide responses to the raised questions and concerns.
>
> **W1:** We agree regarding the specificity of the synthetic tasks, but the complexity of the synthetic tasks is high enough. We could not find any other paper working on arbitrary sorting and retrieval tasks. Usually, similar papers, as cited in our paper, use simple sorting based on alphabetical or numerical natural ordering which can be memorized quickly. The specificity of the tasks is justified by the specific conditions for which indirect attention is designed.
>
> **W2:** We provide the ablation in response to question 1.
>
> **W3:** We provide the response in Question 2 as it is similar.
>
> **Q1:** We have conducted the requested ablation on Pascal VOC dataset and provide the result as follows. The positional bias is $f(P_{ij})$ as the core part of the model.
>
> | Method | Seen classes | Unseen classes |
> |---|---|---|
> | W/O positional bias | 34.3 | 35.27 |
> |W/O dynamic update of positional index | 79.8 | 69.3 |
> | Indirect Attention | 81 | 73.6 |
>
> **Q2:** The orthogonal initialization and unit variance input assumptions in theoretical analysis are a common assumption used in similar theoretical analysis for convenience and tractability, and is also sometimes used in practice as well, especially the unit variance is implemented in the form of data normalization and layer norm. However, in our experiments, we have not limited ourselves to these constraints, and the results we provide are not based on orthogonal weight initialization.
>
> **Q3:** In fact, we think the one-shot objection detection task is a natural and practical example of misalignment where the target image is completely different in size, and objects present in the image from the query image. It is similar to an arbitrary sorting task, where a custom ordering has nothing to do with the given input other than providing a rule for sorting. We agree that application of IA on multimodal (text and image) remains an interesting aspect and a future work.
>
> **Q4:** We provide an ablation in Question 1, but regarding that if the superior performance is from just a simpler model (*though it is a positive point*), the standard self-attention (reported in table 1) has also similar simplicity with the same number of decoder blocks but performs very poorly.
>
> **Q5:** Here is the result of an experiment on a simple synthetic classification task with added noise sampled from $\mathcal{N}(0, I_d)$ to the value sequences only, averaged over 500 data points. Since it is not possible to upload or attach images or PDFs of the curve, we are reporting it in a table.
> | Method | epoch 1 | epoch 5 | epoch 10 | epoch 15 | epoch 20 | epoch 25 | epoch 30 | epoch 35 |
> |---|---|---|---|---|---|---|---|---|
> | Standard attention | 53.6 | 59.7 | 62.4 | 61.8 | 62.6 | 63.7 | 66 | 66.6|
> |Indirect attention | 56.1 | 61.2 | 64 | 66.7 | 68.6 | 71.5 | 72.3 | 74.3 |

---

> > ### Comment · Reviewer_qjrB · 2025-08-05
> >
> > I thank the authors for their response and their clarifications. I suggest including the additional ablations and synthetic experiments with more detail in the appendix of the final draft. This would definitely make the paper stronger.

---

> > > ### Author Response · Authors · 2025-08-05
> > >
> > > We thank the reviewer for their response. In accordance with the reviewer's suggestion, we have included the provided ablation study and the synthetic experiment in the appendix of the paper.

---

### Decision · Program_Chairs · 2025-09-17

**Decision:**

Reject

**Comment:**

This paper investigates "context misalignment" in attention mechanisms, where keys and values are derived from different sequences. The authors provide a theoretical analysis that models this misalignment as structured noise. They then propose a novel mechanism, Indirect Attention, which uses a learnable, adaptive positional bias to function in these misaligned scenarios. The method is evaluated on synthetic tasks and a one-shot object detection (OSOD) benchmark.

The paper has several notable strengths. The theoretical analysis of the proposed problem is rigorous and well-executed. The proposed Indirect Attention mechanism is an interesting extension of traditional attention mechanisms to addressing the specific problem the authors have defined.

Despite the technical execution, there is a fundamental concern about the paper's premise. As noted by Reviewer oBvC, which AC concurred. The motivation and real-world applicability of a setting where keys and values are fully decoupled remain unclear. In most natural applications, keys are intrinsically associated with the values they represent; the paper's core setup feels artificial and does not sufficiently justify why this fundamental association should be broken.

While the authors attempt to motivate this with the OSOD task, it is strange that the query object is used as the key, instead of as the query? In addition, this single application is not enough to overcome the skepticism about the broader relevance of the problem. This concern is amplified by the limited scope of the experiments. The synthetic tasks are explicitly designed to favor the proposed method, and the reliance on one niche real-world task makes it difficult to assess the general utility of the approach. Ultimately, the paper does not provide a convincing argument that this "misalignment" is a widespread, naturally occurring problem that warrants a dedicated architectural solution.

Overall, the fundamental assumption that keys and values should be treated as originating from separate, misaligned sources is counter-intuitive and its applicability is not well-established. Thanks to the authors' effort for submitting the rebuttal, which addresses some of the issues. Yet, the limited and narrow experimental validation does not provide the necessary evidence to justify this unconventional setup.